# Energy Not Exchanged: A Metric to Quantify Energy Resilience in Smart Grids

Hassen Soualah *,†, Gurvan Jodin †, Roman Le Goff Latimier † and Hamid Ben Ahmed †

SATIE, ENS Rennes CNRS, 35170 Bruz, France
* Correspondence: hassen.soualah@ens-rennes.fr
† Current address: ENS Rennes, Avenue Robert Schuman, 35170 Bruz, France.

**Abstract:** In high-impact, low-probability (HILP) events, resilience is defined as the ability of a system to return to a normal operating state after a failure. The generalization of information technologies and distributed renewable production is transforming the power grid into the so-called smart grid, thus allowing for new mitigation methods to address failures. After illustrating the limits of currently existing metrics, this paper proposes a method to quantify the resilience of smart grids during physical line faults while identifying the most impactful failures. For this purpose, a new resilience metric is defined in order to quantify Energy Not Exchanged (ENE). The calculation of this metric in a power grid via the optimal power flow (OPF) serves, therefore, to quantify the extreme resilience of the grid. In addition, various mitigation strategies, which enable maintaining a high level of resilience, despite the presence of failure, are simulated and then compared to one another (tie switch and microgrid formation).

**Keywords:** power grid; high-impact low-probability; smart grid; resilience; microgrid; metric

## 1. Introduction

### 1.1. Failure, Reliability, and Resilience of Power Grids

A power grid is the infrastructure that connects electrical energy production systems to consumers via transmission, distribution, and safety facilities. In the future, this grid will tend to evolve towards the smart grid, and each of its agents will become both a producer and consumer of energy [1]. The capacity of the grid to allow for energy exchange will be challenged whenever faults occur over all or part of it. The sources of faults in the power grid are multiple and imply various consequences. Two quantities provide insight into grid failure.

The first is reliability. In power grids, reliability is defined as the ability of the system to perform a required function under given conditions for a given duration [2]. This concept focuses on the occurrence of failures in a system. The more a system fails, the less likely it is able to function in accordance with its operating conditions, hence the lower its reliability. Reliability is a known and well-documented quantity in the literature [2–5]. Reliability studies are confined to probable and common failures with a limited impact on system performance.

Conversely, resilience focuses on high-impact, low-probability (HILP) events [6]. Such is the case for example of violent storms or earthquakes affecting all or part of a power grid. According to Presidential Policy Directive 21, resilience is defined as the ability of a system to return to a normal operating state following failure [7]. Resilience, however, differs from reliability in its temporal aspect. Unlike reliability, resilience focuses on returning to normal operations, hence the time dependence in defining this quantity. Next, resilience is slightly more subtle in its definition since the emphasis lies not only in qualifying whether a system is functioning, which can be summarized as a binary value, but also in determining a continuous value evolving between the system's total inability to function and its normal

functioning. These differences require models capable of taking into account the temporal evolution of the power system.

### 1.2. Resilience Based on Level of Performance

Resilience can be assessed from the system's level of performance, as shown in the top of Figure 1a. Mathematically, it is determined by the area between the normal case and the trapezoidal shape of the performance level when a fault occurs [8,9]. Several distinct phases can be identified:

- When the system is functioning normally, the performance level is maximal (t < $t_1$);
- When a fault occurs and propagates through the system, then performance gradually degrades ($t_1$ < t < $t_2$);
- When the fault is established, performance stabilizes at a degraded level ($t_2$ < t < $t_3$). This phase corresponds to the time spent to detect the fault and plan restorative actions;
- When system recovery operations are implemented, and the system performance level is restored ($t_3$ < t < $t_4$);
- When system operations are running normally (t > $t_4$).

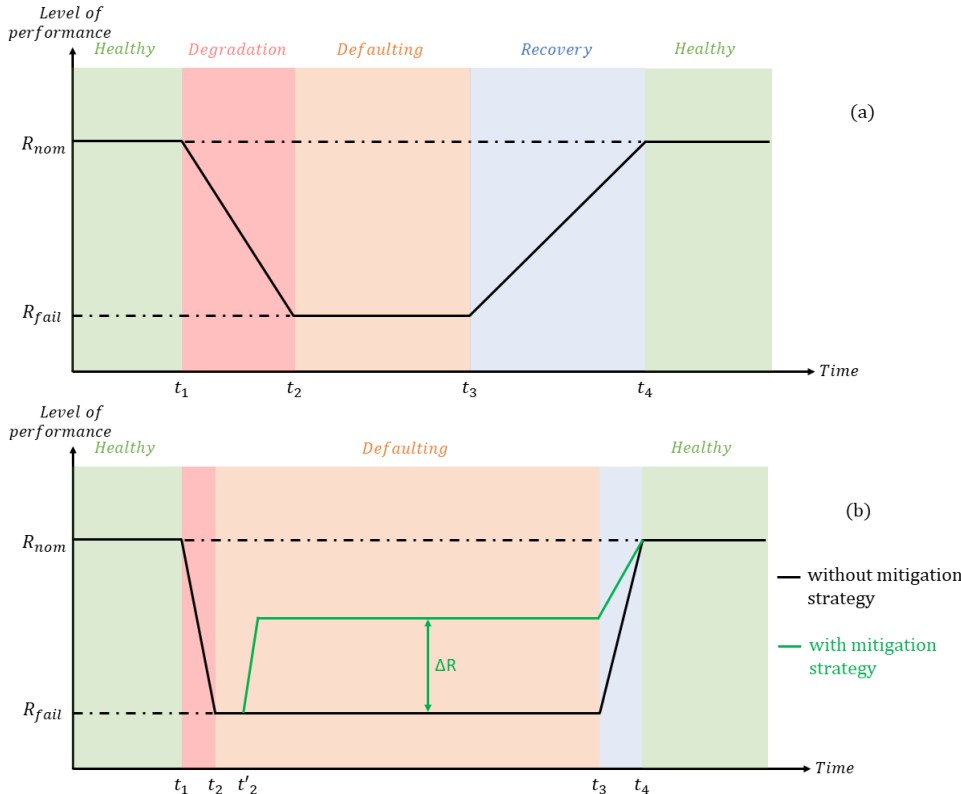

**Figure 1.** Evolution of the level of performance over time: (**a**) the classical shape of the resilience trapezoid; (**b**) the resilience trapezoid, in considering the study hypothesis both with and without a mitigation strategy. The performance starts at its nominal value $R_{nom}$. At $t_1$, a fault occurs, and at $t_2$, the performance stabilizes at a failure level $R_{fail}$. At $t'_2$, mitigation strategy starts, then at $t_3$ the recovery actions commence, and at $t_4$, the system has fully recovered.

Geometrically, this trapezoidal surface is the graphical representation of the responses to four questions summarized by the $\Phi\Lambda E\Pi$ method [9], namely:

- How fast does resilience decrease ($\Phi$)?
- To what level does it fall ($\Lambda$)?
- How long does the degraded period last ($E$)?
- How quickly does the system return to the normal resilience level ($\Pi$)?

Thus, a system is qualified as resilient whenever the fault degrades performance level slowly and weakly; a high performance level is stabilized during faulty operations; the operating time with the fault ongoing is short; and, lastly, system performance returns quickly to its maximum value.

*1.3. State of the Art Regarding Metrics*

The various resilience metrics differ in terms of the measurement they represent. In the case of power grids, the main function consists of guaranteeing supply to the loads; it is common to define resilience metrics related to either the energy consumed by grid loads or, in a complementary manner, the energy not supplied (ENS) in Wh to those loads [6,10–14]. Mathematically, the ENS and the normalized ENS (nENS) definitions are given by Equations (1) and (2).

$$ENS = \int_0^\tau \sum_{i \in \Omega_C} |P_{C_H}^i(t) - P_C^i(t)| dt \tag{1}$$

$$nENS = \frac{\int_0^\tau \sum_{i \in \Omega_C} |P_{C_H}^i(t) - P_C^i(t)| dt}{\int_0^\tau \sum_{i \in \Omega_C} P_{C_H}^i(t) \, dt} \tag{2}$$

where $\Omega_C$ is the space of all grid consumers, $P_{C_H}^i$ is the expected consumed power if there is no failures, $P_C^i$ is the power actually consumed, $\tau$ is the given time of observation of the scenario, and t is the time variable. The normalization is made by the quantity of energy consumed if there is no failure during the $\tau$ duration.

Note that ENS only quantifies load power supply. In a conventional power grid, it makes sense to look only for consumers as this is the main objective of the grid. In smart grids, agents can produce or consume energy; they are called "prosumers". The main objective of smart grids is to ensure bidirectional energy exchanges. Thus, considering only the energy not supplied to the loads misses part of the main objective of smart grids.

It is also possible to define new metrics; the review in [8] lists five desirable properties of any resilience metric:

1. Incorporation of the spatiotemporal variations in the grid;
2. Adequacy of the metric definition in satisfying the study objective;
3. Reliability of the metric in considering the uncertainty of the grid operation;
4. Ease of metric calculation;
5. Feasibility of calculating the metric despite several types of simultaneous failures on the grid.

A given metric cannot necessarily respect all the properties listed above, hence the need to define as many metrics as required.

In order to quantify the defined metrics, three general approaches are followed, by simulation, analytical calculation, and statistical analysis of a failure history [14]. Quantifying metrics by simulation offers the advantage of being easily adaptable. Indeed, it does not require an analytical formulation of the resilience according to grid parameters, nor does it require a failure history on a grid. It is therefore possible to study any grid in the desired scenario as long as its architecture is known. This condition explains why simulation is the most widely used method for quantifying resilience [14]. To calculate the resilience metrics, it is necessary to

- Define the power grids to be studied;
- Assign to each agent of the grid the powers being exchanged;
- Select or draw random failures;
- Calculate the values required for the resilience metrics.

Because of the plurality of problems involved, the studies have differed in size (from a few nodes to several dozen), type of grid simulated (distribution grid with a radial architecture or transmission grid with a meshed architecture), nature of the faults (prob-

abilistic or deterministic), and, lastly, simulation duration (from a few hours to several hundred) [6,10–14].

### 1.4. Resilience Studies

This section highlights the resilience studies exposed in the literature. In power grids, work has been limited to the study of failures defined upstream under different scenarios that are more or less critical [6,13,14]. The power grid restoration procedures are predefined and simulated. These studies do not cover all possible line failures on the considered power grid thanks to a preliminary study conducted using human expertise; they therefore do not identify the most deleterious cases that could possibly occur according to an automated method. The selected powers exchanged are static, hence the variability in production and consumption is not taken into account, which means that depending on the exchanged powers, the resilience can fluctuate. The identification of the worst cases is a key point when discussing resilience and even more so for HILP failures. Moreover, although [14] deals with the addition of tie switches and a microgrid formation on the grid, here, again, arbitrarily chosen scenarios are being simulated. The literature thus misses a point in defining a method that systemically studies resilience for all HILP failures. As previously mentioned, resilience metrics are defined around the ENS of the load. However, for a smart grid, since every power grid agent can be a producer and a consumer, the need exists to define a new metric that conveys the capacity of the smart grid to allow all agents to exchange as much energy as needed.

### 1.5. Scope, Contribution, and Limitations

The work presented below has multiple objectives. The first one is to define a new metric to assess the resilience in smart grid. The second one is to define a generic method to assess the proposed metric. The third one is to compare the results from ENS and the new metric proposed. The last one is to compare hardening and operation mitigation strategies to enhance resilience (see Section 2.4). For this purpose, the approach is based on simulations in order to gather enough data to perform relevant analyses. Resilience indices of baseline simulation cases are compared to cases with failures featuring leverage actions (e.g., tie switches, flexibility, microgrid formation) that will be simulated by optimal power flow (OPF), and the results will be discussed. The work carried out will focus on the distribution power grid, given that this part of the grid houses the distributed energy generators and especially the renewable ones. For example, in France, the photovoltaic power installed in the distribution power grid increased from 0.3 GW in 2010 to 13.3 GW in 2022, according to Enedis, the main distribution system operator (DSO) [15].

Concerning the resilience trapezoid, the entire shape will not be assessed by making any assumptions. In this study, only one line will be down at any given time. Thus, the degradation time of the fault is nearly instantaneous, as is the recovery time (Figure 1). The choice to study line failures is justified by the high impact of these failures on the grid. In fact, one failed line can disconnect multiple agents on the power grid and then have a great influence on resilience study. Nevertheless, rare events such as natural disasters often cut more than one line. This is a limitation of the present study, but the proposed approach can be simply adapted to cases with several simultaneous failures. Since the goal of this study is to assess the resilience provided by mitigation strategies, fault detection and isolation are considered to have been correctly performed by the DSO. In addition, the duration $t_2' - t_2$ denoting the time to activate the resilience mitigation strategies is considered short, and thus negligible relative to the fault duration. According to these hypotheses, the trapezoidal shape of resilience is similar to a rectangle (Figure 1). The only outstanding value to be calculated is $R_{fail}$.

The remainder of this article is organized as follows. Section 2 presents the new resilience metric defined in the smart grid, along with the grid under study and the data, tools, and methodology used to evaluate the metric. Section 3 details the various simulated

cases, while Section 4 provides the set of simulation results. Section 5 discusses theses results, and a conclusion of the work is offered in Section 6.

## 2. Materials and Methods

### 2.1. Defining a New Metric of Resilience of Smart Grids

Current resilience metrics are primarily based on the ENS to the loads. This is justified because the main purpose of the power grid is to supply loads. This metric is limited in its consideration of smart grids because power exchanges are bidirectional, but the definition of a new metric could be inspired from the ENS metric. As a smart grid's main objective is to guarantee bidirectional power exchange between agents, the new metric must depend on the power of prosumers and not only consumers. In addition, the metric must comply with the list of desirable properties described in Section 1.3, including the ease of calculating the metric, the pertinence of its definition, etc. In light of these constraints, the energy not exchanged (ENE) is defined as the sum of the difference between expected power and power actually exchanged by all agents in Equation (3) and the normalized version nENE in Equation (4):

$$ENE = \int_0^\tau \sum_{i \in \Omega} |P_{G_H}^i(t) - P_G^i(t)| + |P_{C_H}^i(t) - P_C^i(t)|dt \tag{3}$$

$$nENE = \frac{\int_0^\tau \sum_{i \in \Omega} |P_{G_H}^i(t) - P_G^i(t)| + |P_{C_H}^i(t) - P_C^i(t)|dt}{\int_0^\tau \sum_{i \in \Omega} |P_{G_H}^i(t)| + |P_{C_H}^i(t)|dt} \tag{4}$$

where $P_G^i$ is the power generated by agent i, $P_C^i$ the power consumed by agent i, and $\Omega$ is the space of all the agent except for the point of common coupling (PCC). The PCC is voluntarily excluded from the agent space since it is considered to be a balancing agent guaranteeing the power balance in the linked grid. The ENE is normalized by a reference scenario, referred to as the healthy case with a $\tau$ duration; this case considers the power exchanged provided that no failure occurs on the grid named the healthy case. The quantities indexed H are those. The duration of failures studied in the given scenario cannot exceed the $\tau$ duration. All power consumed or produced is taken as positive.

Should the failure disable the entire ability of the grid to exchange power, then the nENE will equal 1. Conversely, if the failure has no impact on the exchanged power, nENE will equal 0. Therefore, nENE $\in$ [0;1].

Regarding the five desirable properties of the resilience metric indicated in Section 1.3, ENE respects all of them, which in turn guarantees the correct definition of ENE as a metric of resilience with respect to [8]. Indeed, the ENE definition is simple and focused on smart grids, i.e., those being studied herein. In addition, the power exchanged by the agent on the grid will be influenced by the line failure. Indeed, a line in failure modifies the power exchanged, which means that it will logically impact ENE. To ensure a reliable value of the metric when considering grid operations, the power exchanged by the agent will be modified each hour during the simulation for the whole one-year database, thus, giving results taking into account the seasonal variability of the data. Therefore, mean and dispersion of the metric will be reliable. Further details will be provided in the following sections. Lastly, it is possible to take into account multiple failures depending on the simulation. This respects the last point of Section 1.3, imposing the feasibility of calculating the metric with the consideration of simultaneous failures.

Moreover, it is important to notice that ENE directly derives from ENS. Indeed, ENE assesses the amount of energy not exchanged by the prosumer agents of the smart grid. Nevertheless, if at a given time, all agents did not produce any power to the grid, the sum of $P_G^i$ will be equal to 0 and only the terms $P_C^i$ will remain (3). ENE will be strictly equal to ENS. Thus, ENE appears to be an extension of ENS in power grids, to the smart grids allowing bidirectional power exchanges. Later in the paper, Section 4.1 details a comparison between ENS and ENE in order to discuss why ENE is more suitable than ENS

to assess resilience in smart grids. Finally, ENE is a better metric to optimize flexibilities provided by a smart grid than ENS.

### 2.2. Studied Grid and Database

To respect the scope of this study, the chosen grid needed to be a distribution power grid. Ideally, this grid is defined both geographically and technically. In addition, the power exchanged should include renewable energy production. As such, a low-voltage distribution grid from CIGRE [16] was chosen; such a grid is typically found in neighboring European countries. The choice of this specific grid is arbitrary; other grids from IEEE could have been selected as well. The architecture of the grid is illustrated in Figure 2, where 15 prosumers are distributed over the 40 buses of the three-phased 400 V line-to-line power grid. The PCC is represented by the hatched square on top of the grid. All agents except the PCC lie below the 20 kV/400 V transformers. Agents are distributed into groups R, I, and C, each of which is supplied by a single transformer; they are placed at the extremity of each branch of the grid. Groups R and C are similar, unlike Group I, which is composed of just one isolated agent quite distant from the PCC.

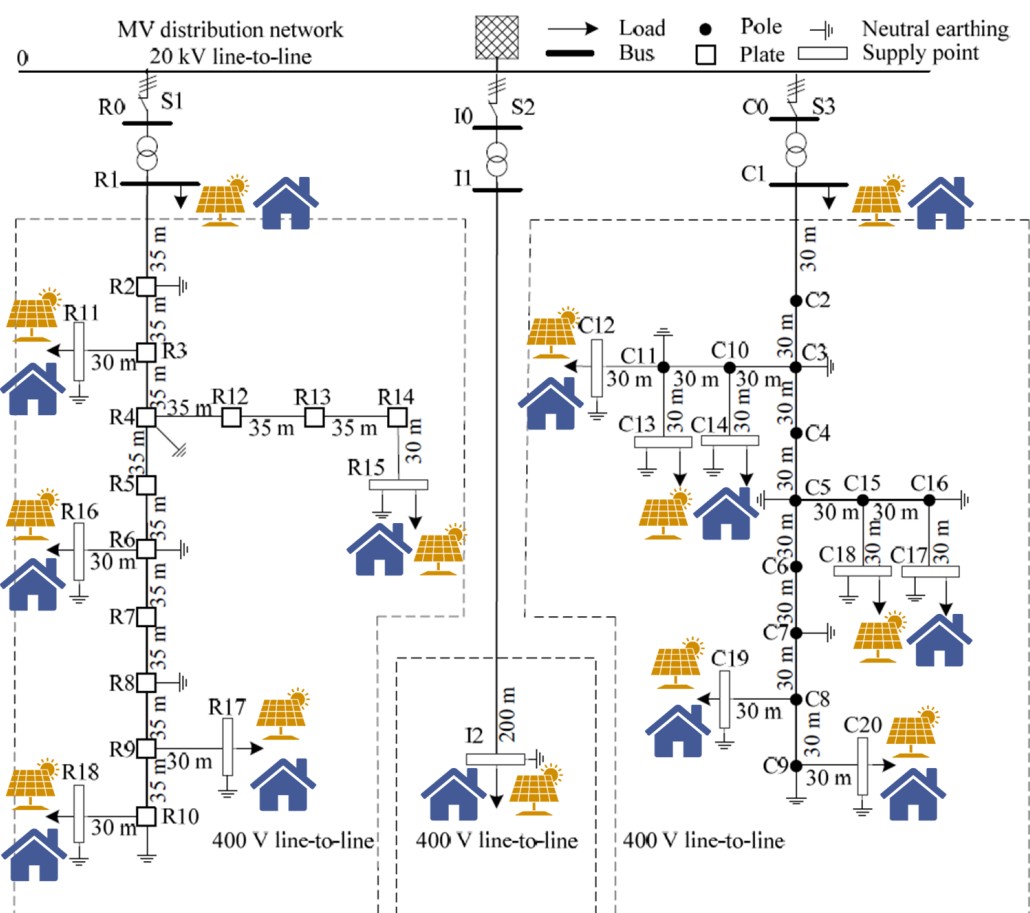

**Figure 2.** Grid architecture. The blue houses represent residential consumption, and the yellow solar panels denote solar production. These symbols are placed at each arrow as a representation of loads in the grid. The PCC is depicted by the hatched square on top of the grid.

All data required to simulate the grid are available in [16]. No changes were made other than the consumed and produced powers. The related databases used were UMASS Smart* Dataset (2017) and Solar TK; they were created at the University of Massachusetts and detailed in [17]. The simulations were run over a time span of one year in order to include all seasons with their specific consumptions and productions. Figure 3 shows a sample of the database.

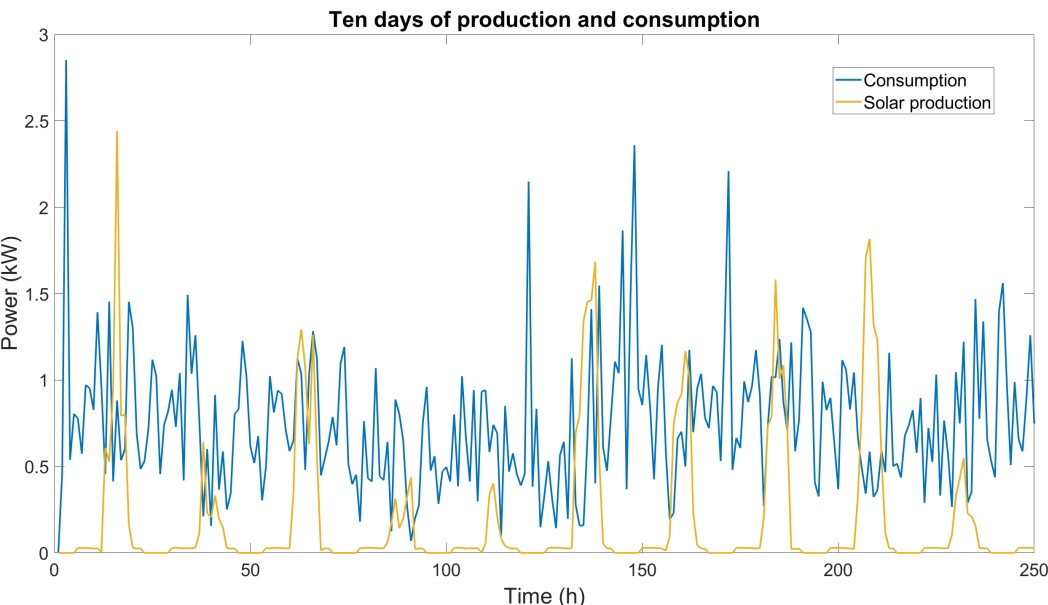

**Figure 3.** Ten days of solar production (yellow line) and consumption (blue line).

*2.3. Simulation*

2.3.1. Optimal Power Flow (OPF)

To simulate power grid operations, Matpower was used [18]. This toolbox is an extension of Matlab and implements functions to solve OPF-type problems. The power grid is composed of a set of nodes N linked by a set of lines L. Inside the nodes, a set of generators G and consumers C exchange active and reactive power P and Q. The nodes have a voltage magnitude V and angle $\theta$. Solving an OPF implies solving a constraint minimization problem. With the sets defined, this minimization problem can be written with the sum of the cost function, f, of all agents to be minimized (5) subject to the equality constraint due to the balance of apparent power in each node (6), Ohm's law linking the apparent power flowing between two nodes i and j across an admittance Y at voltage magnitude V (7), within the limit of apparent power flow in the lines (8) and a variable limit inequality on V, $\theta$, P, and Q that constrains them between their minimal and maximal values (9)–(12) [18]:

$$\min_{V,\theta,P,Q} \sum_{i \in N} f_P(P_G^i) + f_Q(Q_G^i) \tag{5}$$

subject to:

$$S_G^i - S_C^i = \sum_{(i,j) \in E} S_{ij} \quad \forall i \in N \tag{6}$$

$$S_{ij} = Y_{ij}^* V_i V_i^* - Y_{ij}^* V_i V_j^* \quad \forall (i,j) \in E \tag{7}$$

$$|S_{line}^i| \leq |S_{line_{max}}^i| \quad \forall i \in L \tag{8}$$

$$|V_{min}^i| \leq |V^i| \leq |V_{max}^i| \quad \forall i \in N \tag{9}$$

$$\theta_{min}^i \leq \theta^i \leq \theta_{max}^i \quad \forall i \in N \tag{10}$$

$$P_{min}^i \leq P^i \leq P_{max}^i \quad \forall i \in G \tag{11}$$

$$Q^i_{min} \leq Q^i \leq Q^i_{max} \quad \forall i \in G \tag{12}$$

$E$ is a matrix representing the linking of the nodes. If a line exists between nodes i and j, then the value $E_{ij}$ is the couple $V_i$, $V_j$, and 0 otherwise.

### 2.3.2. Constraints

The constraints considered in voltage magnitude are [0.9;1.1] p.u. As a consequence of the study and therefore the degraded power grid operations, the voltage angle limits are [−360;360] degrees to ensure convergence of the OPF. The active and reactive power consumed lie between 0 and the value read from the corresponding database. For producers, the database provides the maximum value of active power produced, with the minimum once again here set to zero. The absolute reactive power exchange is equal to the produced active power, which means that the inverter in the solar installation can produce or consume reactive power to support the grid in regulating the voltage magnitude. All other constraints (e.g., lines, PCC power) were specified in the paper defining the grid [16].

### 2.3.3. Cost Functions

In the simulation, the objective is to encourage the OPF to minimize the sum of the cost function with respect to the power grid constraints. Moreover, the OPF must allow the consumer to consume exactly $P_{ref}$, and the producers to maximize their production within the limit of producible power, with PCC establishing the balance. $P_{ref}$ reflects the power specified by the database. In order to proceed, a zero cost will be assigned to solar generation, with a cost function slightly curved and centered at the origin for the PCC and a strongly curved function in front of the PCC cost function, for the consumers centered in $P_{ref}$, as shown in Figure 4. Consumers are specified in Matpower as a constant power factor load, meaning that the reactive power demanded by each consumer follows the active power demand with the same cost function. Each cost function included can be written as a second-degree polynomial equation depending on the active power exchanged P (13):

$$f(P) = a_2 P^2 + a_1 P + a_0 \tag{13}$$

The three parameters $a_2, a_1$, and $a_0$ are computed to have a positive cost function with a minimum at the power specified by the database equal to 0. In order to compute the parameters $a_2, a_1$, and $a_0$ expected in Matpower, they are identified with the polynomial canonical form that is simpler to design (14).

$$f(P) = \gamma(P - \alpha)^2 + \beta \tag{14}$$

where $\gamma$ is the curvature of the polynomial function, with $(\alpha, \beta)$ the coordinates of the minimum. By definition, the minimum is located at the point $(P_{ref}, 0)$. From identification, $(\alpha, \beta)$ equals $(P_{ref}, 0)$. To ensure the desired behavior of the PCC, consumers, and solar generators, $\gamma$ will be, respectively, 100, 1000 and 0.

Moreover, the values $a_2, a_1$, and $a_0$ are found by developing (14) and identifying its expression with (13). The three equations linking $a_2, a_1, a_0, \alpha, \beta$, and $\gamma$ are listed below (15)–(17):

$$a_2 = \gamma \tag{15}$$

$$a_1 = -2\gamma\alpha \tag{16}$$

$$a_0 = \gamma\alpha^2 \tag{17}$$

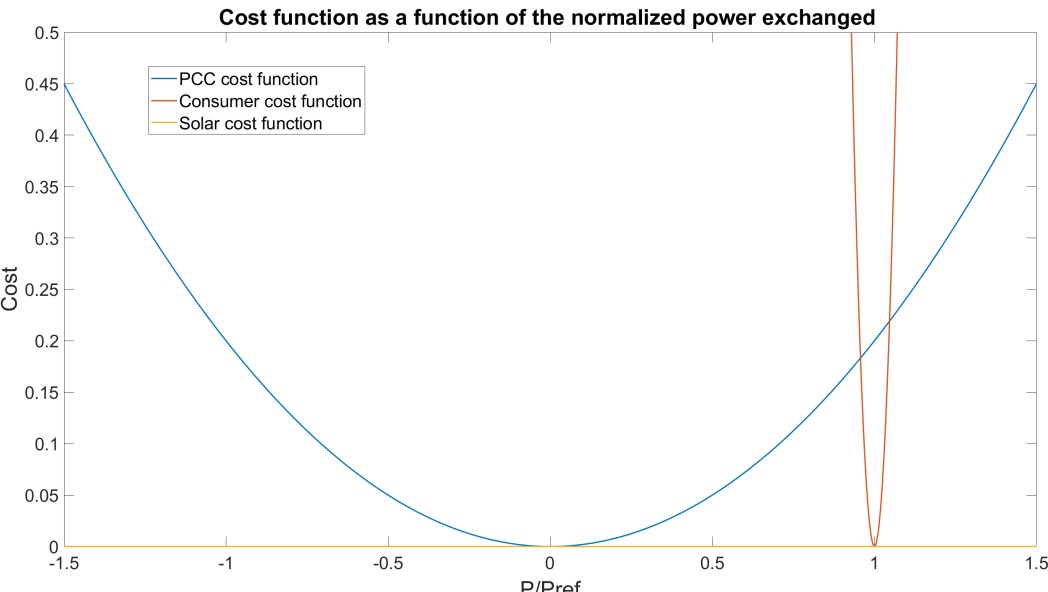

**Figure 4.** Cost function associated with the agents of the grid.

### 2.4. Mitigation Strategies

In [9], the measures intended to increase resilience are combined into two groups, namely:

- The hardening resilience measure, which boosts structural resilience by undergrounding lines, adding redundancy in the lines, using robust materials to construct the poles supporting the lines, etc. These solutions are more efficient in increasing resilience, yet are less cost-effective than the operational measures [19].
- Operational measures, which are based on grid management with regard to forecasting and decisional tools.

Due to the plurality of the strategies existing to enhance resilience [9], the present study only addresses two of them: tie switches and microgrid formation. This choice was made because other studies are currently using these strategies [6,10,14]. Furthermore, the smart grid is designed with an evolving architecture, hence the strategies fit well with this paradigm. These solutions will be described below. Let us recall that in this study, the detection and isolation of the fault, the activation of the measure, and the stability of the grid are all out of range; it is therefore considered to be correctly determined by the DSO.

#### 2.4.1. Tie Switches

Tie switches are lines installed but not used in the grid. When a fault occurs, a manual or automatic operation can close the connection between the line and the grid. Tie switches create a new path for the power flow. The most important fact here is that the switches can modify the radial architecture of the distribution grid into a meshed architecture closer to the transmission grid. Thanks to the new power flow path, the voltage amplitude at each node may be impacted. Two grids simulated via OPF, possibly including tie switches, lead to completely opposite results. Thus, this solution exerts a structural impact on the resilience by opposing in the disconnection of agents (Figure 5).

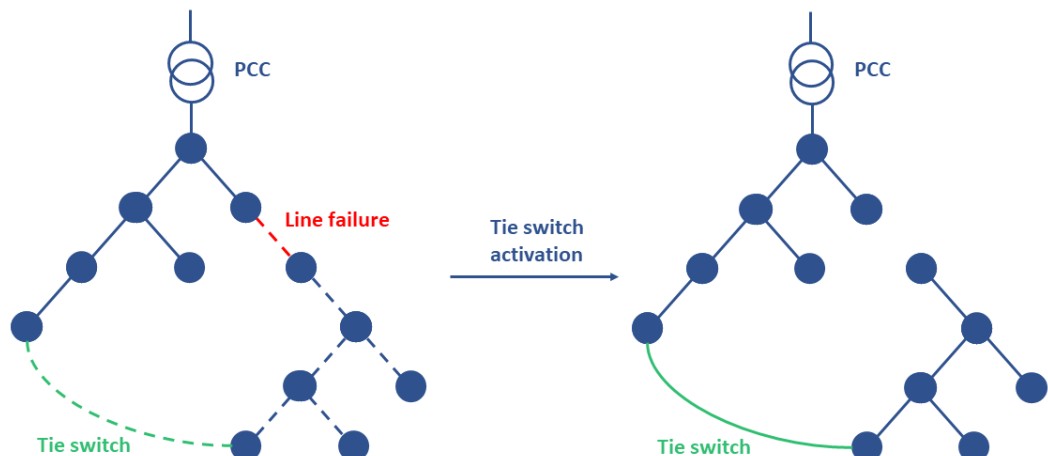

**Figure 5.** Tie switch activation in the case of line failure in a distribution grid—the dashed lines represent a power flow interruption.

### 2.4.2. Microgrid Formation

In contrast, this strategy enhances resilience after the disconnection of agents. By associating all possible disconnected agents into a microgrid and allowing them to exchange the local power produced, the microgrid increases resilience operationally (Figure 6). This mitigation strategy is sensitive to the quantity of local power produced. The more power produced, the more the microgrid tends to exchange the same amount of power as in the healthy case.

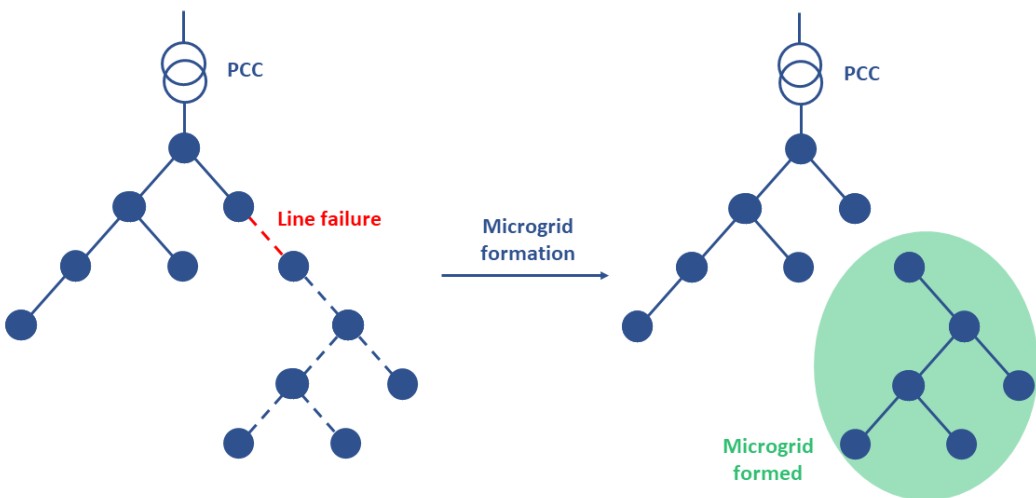

**Figure 6.** Microgrid activation in the case of line failure in a distribution grid—the dashed lines represent a power flow interruption.

### 2.5. Method

The main goal of this article is to define the value $\Delta R$ gained by adding mitigation strategies (Figure 1). In pursuit of this goal, Figure 7 summarizes the general approach to calculating the metric of resilience, as explained in Section 1.3. After database and grid loading, one line failure is selected. For this failure, the entire database is simulated for a given mitigation strategy. This process is repeated until all time steps for all line failures have been simulated. Since the whole year is being simulated, the best and worst cases of resilience can be identified. The simulations are conducted on a computer with an AMD Ryzen 7 3700x 8-core, 3.6 GHz CPU. Simulating the entire database for all 40 possible line failures takes roughly 3 h and 20 min.

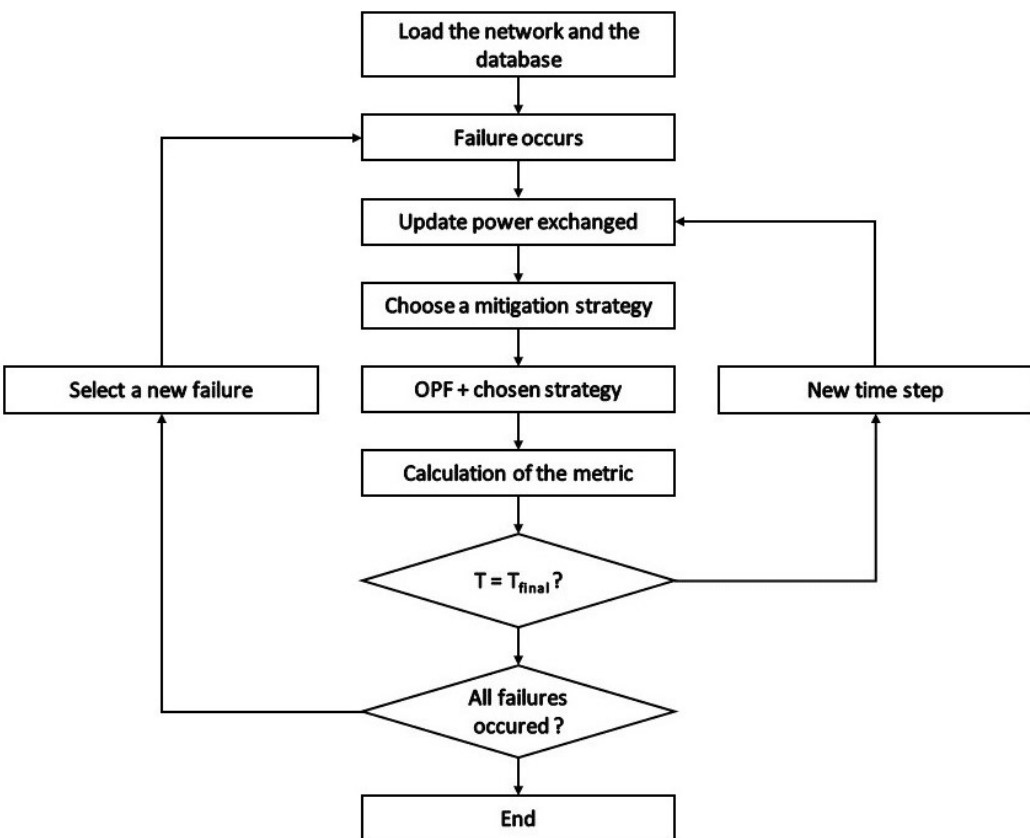

**Figure 7.** Algorithmic approach to quantifying resilience in the presence of failure.

### 3. Cases

The value of $\Delta R$ in Figure 1 is based on the performance assessed by ENE. Because ENE has been normalized by the healthy case, a failure-free case is needed. The dark line represents the performance of the power grid without a resiliency-enhancing strategy in the presence of a fault. Each mitigation strategy will be simulated separately. The final case will simulate both strategies in order to calculate the best attainable $\Delta R$ value. A more detailed description of this case will be given in the following subsections.

#### 3.1. Healthy Case

According to this case, the power exchanged (thus, the ENE) is evaluated at each point of the database, which serves to define the reference for calculating ENE. By definition, since this case normalizes ENE (3), the value of ENE for all database points is equal to 0.

#### 3.2. Fault Case

Here, the power grid is being simulated at each point for each possible line failure; the ENE is then calculated. The subnetwork located below the line in fault is not simulated due to the absence of a mitigation strategy to enable the power exchange. The level of performance obtained is the worst possible.

#### 3.3. Fault Case with the Best Tie Switch Case

Given the existence of 40 buses, 780 potential tie switches are able to link the combinations of two different buses; however, all switches are not useful. The aim here is to add the best possible tie switch based on a preliminary study. For the sake of expediency, all combinations will not be simulated. Indeed, if a tie switch links two juxtaposed buses, it will prove to be inefficient because the remainder of the grid will be unchanged and still sensitive to failure; this explains why line failure is equiprobable. Therefore, all lines linking two juxtaposed buses will not be simulated. The distribution power grid is radial,

arborescent, and separated into three areas (Figure 2). In the case of line failure, agents located in the branch extremities will undergo a power interruption. The best way to ensure a continuous power supply to those agents consists of linking them to one of the branch extremities. The further downstream this connection takes place, the greater the number of agents that will be supplied by the tie switch. The left and right areas of the network are those with the most agents. Linking extremity agents in different areas of the grid tie switch should yield the best increase in ENE. The identification of the best tie switch is given in Section 4.

*3.4. Fault Case with Microgrid Formation*

When a line failure occurs, the subnetwork isolates from the PCC (Figure 6). As opposed to the failure case, the subnetwork here will allow the agents to exchange power within the limits of the locally produced power. By virtue of solar production, whose main characteristic is variability, nothing guarantees that all agents will consume and produce the power $P_{ref}$ that they seek. Three possible situations may arise:

- The power produced and that is consumed in the subnetwork are strictly equal. In this situation, the power exchanged will yield the same value as in the normal case. The ENE will thus equal 0;
- The power produced is less than the power consumed in the subnetwork. Consumers must accept to be flexible and lower their consumption. The ENE will be strictly greater than 0;
- The power produced is greater than that consumed in the subnetwork. Producers are unable to sell all the power produced to consumers in order to respect power balance. They therefore need to cut production to the amount of power demanded. Again, the ENE will be greater than 0.

Nevertheless, the mean ENE in this case cannot be higher than the fault case.

*3.5. Fault Case with the Best Tie Switch and Microgrid Formation*

This last case simulates both mitigation strategies. The objective here is to show that with these strategies, the power grid is extremely resilient to line failure. The tie switch avoids disconnecting agents from the grid, while the microgrid formation increases resilience in those few cases when disconnection occurs.

## 4. Results

*4.1. Comparison between ENE and ENS*

In Section 2.1, ENE is defined and justified for smart grids as the ENS in power grids. As a reminder, ENE is equal to ENS only in cases where renewable production is zero. In this study case, it happens especially when it is the night, about the half of the database points. Let us focus on the case where microgrid formation is allowed for agents disconnected from the PCC and a failure appears during the daytime. In this case, comparing ENS and ENE, it is interesting to see the difference between them. For the whole database, the fault duration $\tau$ varies from 1 to 24 h to have the influence of the daytime. The self-producing index (SPI), i.e., the amount of energy produced in the year, is twice the power consumed in the smart grid. The results are presented on the top of Figure 8.

First of all, ENE and ENS follow the same trend and increase linearly with the fault duration. Means are quite similar, too, even if there is a slight difference between them; on average, ENE is larger than ENS. The main difference lies in the minimum. ENS can be equal to zero, unlike to ENE. This can be explained by the consideration of the power produced in ENE. More particularly, distributed solar producers must cut off their production at the level of consumption to respect the power balance in the formed microgrids. Indeed, in the microgrids, the surplus of solar production cannot be absorbed by the PCC because there is no connection with it. This case is illustrated in Figure 9.

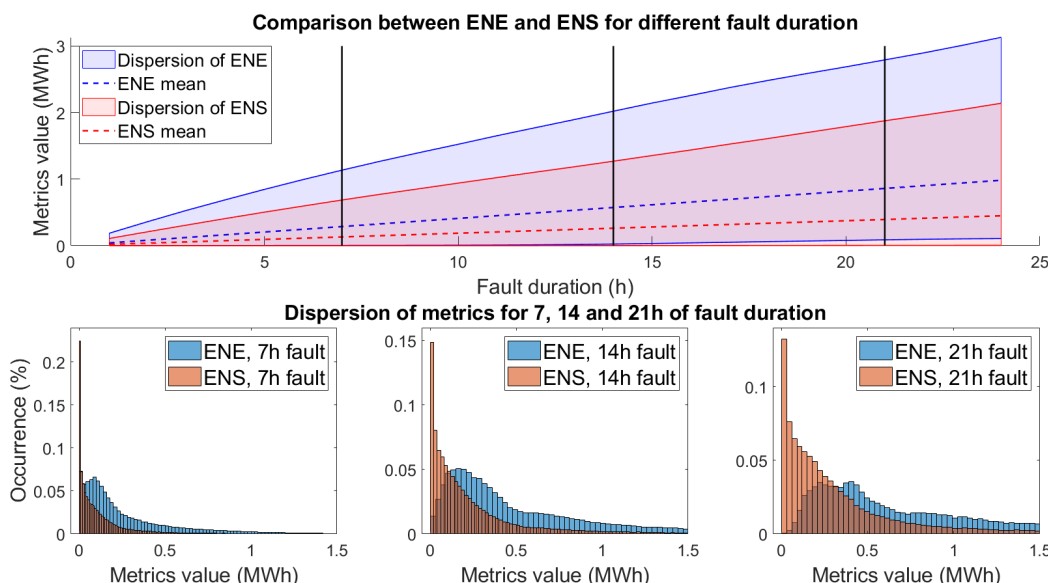

**Figure 8.** Comparison of the metrics. The results are from the simulation of the case considering SPI = 2, microgrid formation is allowed, dataset for 1 year. Top: comparison between ENE and ENS for different fault duration. Colored area represents the dispersion of all the values taken by the metrics. Dashed lines: their means. Bottom: for three values of fault duration (7, 14, and 21 h), dispersion of metrics are compared by their histograms.

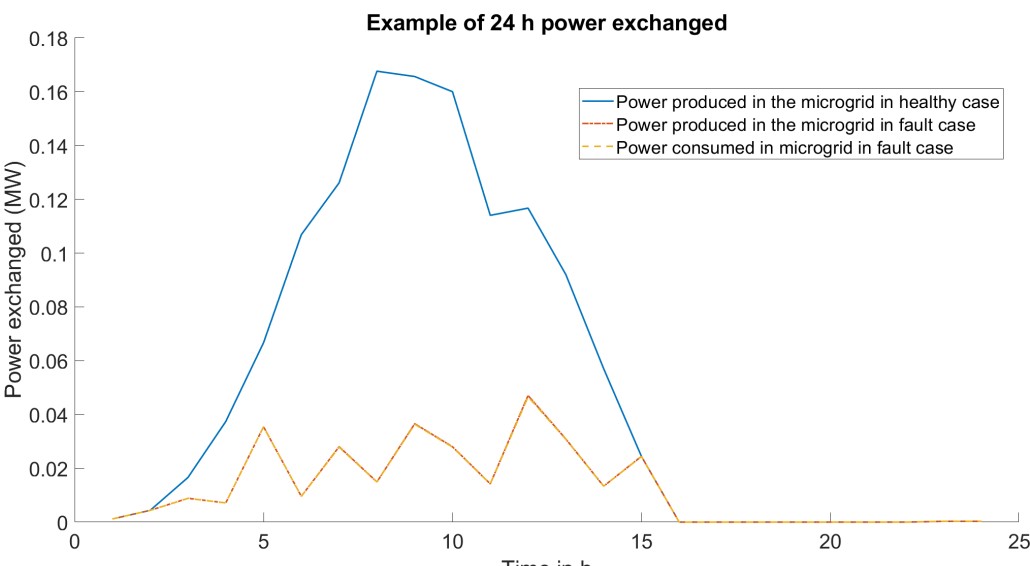

**Figure 9.** Example of 24 h power-exchanged smart grid with a microgrid formed by nodes from R2 to R18. The line linking nodes R1 and R2 is in failure.

The bottom of Figure 8 shows histogram representation of the values taken by the metrics for a specified fault time duration. At a given fault time, dispersion of ENE and ENS are different. ENS dispersion looks similar to a hyperbole, unlike the more spread-out dispersion of ENE. The occurrence expressed in percent means that for a given fault duration, such percentage of the metric is comprised of the bin width of the histogram. This difference is explained here again by the definition of the metrics. By considering only consumers, there is greater chance to obtain 0 MWh of energy not supplied than by considering all prosumers of the smart grid. For both parameters, increasing the duration of failure spreads the dispersion to higher values, but the mean value of ENE distribution is more sensitive to the fault duration time.

For this case, ENS is often equal to 0 because there is enough local production to balance the consumption. If only ENS is considered, the power grid is at its maximal resilience because all consumers are fully supplied. However, if the metric used is ENE, the interpretation of the results change. Solar production is cut off to the level of the consumption (Figure 9). Therefore, ENE is strictly greater than 0. Thus the smart grid is not at the maximal resilience level. The more the solar production, the more ENE will be high.

In conclusion, ENS is designed to assess the performance of a common power grid. When this metric is used in smart grids, it fails to correctly represent the level of performance of the grid because agents are not only consumers but prosumers. Therefore, the metric ENE is needed to extend the notion of energy not supplied to all the prosumer agents. Thus, in what follows, only the ENEs are plotted and discussed.

### 4.2. Study of the Grid in Presence of Resilience Mitigation Strategies

The addition of a tie switch is the first mitigation strategy considered. The best tie switch to add, in consideration of nENE, lies between nodes R18 and C20 (Figure 2). Other good tie switches link nodes R18 and C19, and R18 and C20. As explained in Section 3.3, the best tie switch links nodes located at branch extremities in the downstream part of the power grid. It is also important to note that the two nodes linked are from different areas. The fault case with the best tie switch presented includes this additional line. The other mitigation strategies that are simulated are fault case with microgrid formation, and the combination of the best tie switch with microgrid formation.

The simulation compared all cases individually with a self-production index (SPI) equal to 0.5 (Figure 10), which implies that during the simulated year, solar generators produce half the energy consumed. The blue lines indicate 95% of the nENE values over the year. The black squares indicate the mean nENE for each case. The yellow shadows show the nENE probability density. The higher the resilience, the denser the probability that it lies around 0. Thus, an nENE equal to 0 means that the amount of energy exchanged equals the energy exchanged in the healthy case. As a consequence of its definition, the healthy case equals 0 for each hour, which explains the lack of dispersion for this case and a mean value of 0.

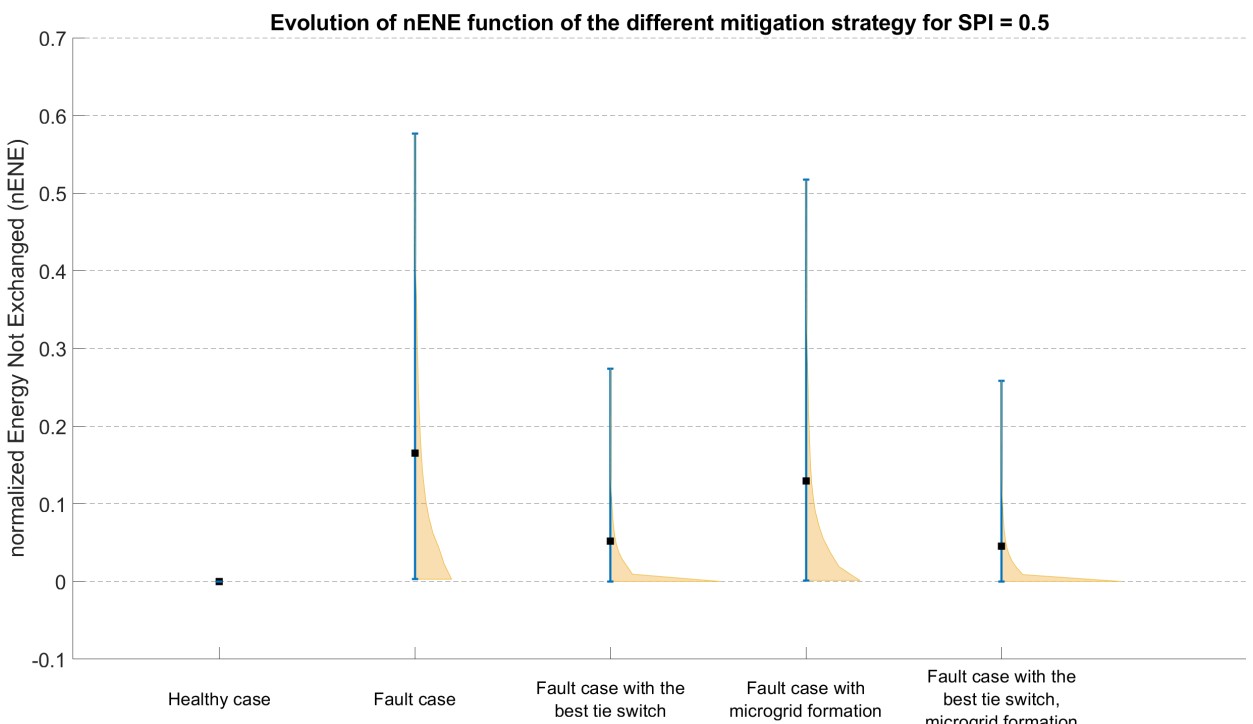

**Figure 10.** nENE comparison for the simulated cases with the different mitigation strategies. The blue lines reflect 95% of the most representative values established by nENE over the 1-year dataset. The black square represents the mean nENE, and the yellow shadow depicts the nENE probability density.

In Figure 10, for the fault case, the nENE is $\in$ [0.01; 0.62]. Thus, in the worst case, a line failure can prevent around 60% of the energy from being exchanged. In all other cases, mitigation strategies increase the value of nENE. Adding the best tie switch to the power grid has three effects:

1.  Reducing extent of dispersion by a factor of 2;
2.  Reducing the worst impact of a fault line by 30%;
3.  Concentrating the density of nENE around 0.

Indeed, the percentage of nENE values lower than 0.05 decreases from 50% in the fault case to 9% in the tie switch case. Note that in the best case, the nENE equals 0.

Microgrid formation proves to be less effective than adding a tie switch. Indeed, the dispersion is reduced and the nENE density probability shifts closer to 0, yet not as efficiently as in the previous case: only 60% of nENE values are below 0.05. Using both mitigation strategies provides nearly the same results as those of the tie switch. The number of nENE values lower than 0.05 increases to 92% in this case vs. 89% in the tie switch case. In conclusion, for this figure, choosing the best tie switch to add into the power grid, and allow microgrid formation, is the best choice to increase resilience.

As noted in Section 3.4, for the two cases simulating microgrid formation, the solar power generated is clearly correlated with the nENE value. In order to discuss the results regarding the solar power generated and thus the SPI value, Figure 11 shows the value of nENE for various values of SPI derived from the simulated cases. For each SPI value, a new healthy case is obviously defined. Figure 11 indicates that the SPI value does significantly influence the nENE dispersion when its value increases, especially for the low values. Above the value 0.5, increasing SPI has almost no effect on the nENE dispersion. In the fault case with microgrid formation, when SPI is equal to 0, the results are the same as the fault case because the microgrid cannot work without local production.

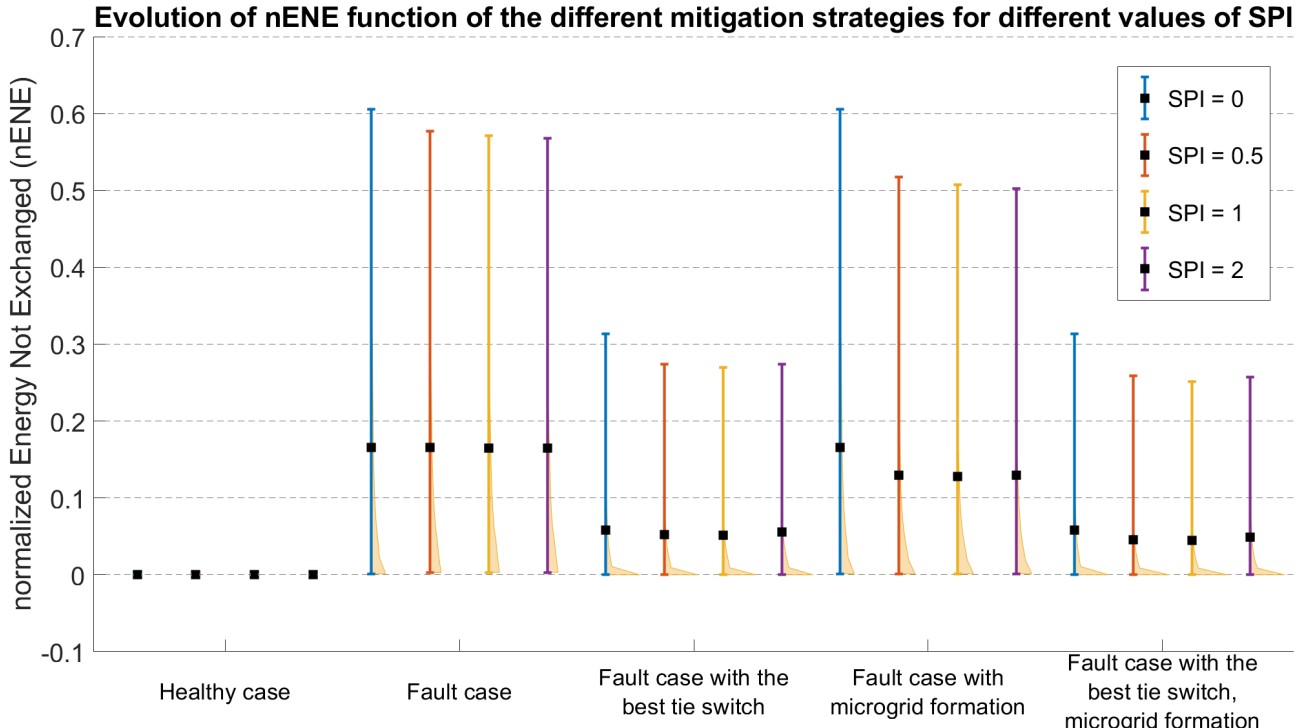

**Figure 11.** nENE comparison for the various simulated cases and self-production index values. The blue, red, and yellow lines reflect 95% of the most representative values established by nENE over the year. The black square represents the mean nENE over the year, and the yellow shadow depicts the nENE probability density.

## 5. Discussion

These results reveal some interesting findings. First, the tie switch has a distinct impact on the resilience of the power grid; its simulation was found to be the best. This outcome is explained by the fact that with this strategy, agents in the majority of cases are always being supplied with power. Second, allowing for microgrid formation and agent flexibility is less efficient than with the tie switch, which can be explained by the nighttime values reaching half the database values. Third, from an nENE perspective, it is preferable to keep agents linked to the PCC most of the time. Fourth, from a DSO point of view, this grid does not need to incentivize the quantity of local solar production above an SPI value of 0.5. Above 0.5, SPI seems to exert almost no influence on nENE should the agents be linked to the PCC due to the fact that the PCC is designed to supply all agents in the grid. Thus, when agents are linked by means of a tie switch, all solar power will be injected into the grid regardless of the power produced. Fourth, in cases including microgrid formation, the slight improvement in nENE is explained by the decorrelation between production and consumption. Indeed, when agents are linked to the PCC, power production is fully correlated with consumption since every agent can be supplied. However, in cases with a microgrid, these two quantities are no longer correlated, hence the power supply falls short of that when the microgrid is connected to the PCC. To ensure the correlation between production and consumption, adding an energy storage for the agents might be a solution worth studying.

While these results are only valid for this grid, they do offer an initial approach to quantifying the best strategies to enhance distribution grid resilience (tie switch and microgrid formation). The grid studied herein has a mean nENE value equal to 0.17 in the fault case, which is quite low. This same study in another grid may lead to different results and interpretations.

## 6. Conclusions

This work proposes a method to quantify the resilience in a power grid that has been tailored for smart grids. All phases of the resilience trapezoid were not assessed, only the faulty one. Because smart grids differ from conventional grids with regard to bidirectional power exchanges, resilience cannot be assessed by the same typical metrics as ENS. A new metric was thus defined to represent the level of performance of a smart grid, denoted energy not exchanged (ENE). This ENE was then assessed in a distribution power grid in integrating solar generation and various resilience mitigation strategies, such as tie switch and microgrid formation. For the chosen distribution grid, the best strategy appears to be the tie switch, relative to both the ENE mean and its dispersion. These results also revealed a high impact of the local generated solar power on ENE, especially for values of self-producing index (SPI) equal to 0.5. Above this value, the effect is mitigated. All results shared are only valid for the grid being studied.

This work constitutes the first step of a more complex study. Certain hypotheses and arbitrary choices were made, such as the selected grid and database, or with the choice of only one line failing at a time. Such a setup excludes the possibility of a total loss in the connection between the grid and the PCC. In an upcoming study, it will be interesting to focus on a more degraded operation with several faulty lines. The best resilience mitigation strategies can evolve because of the absence of PCC.

Lastly, this study was not comprehensive. Other strategies are able to simulate and raise discussion on, for example, energy storage, which can enhance the correlation between production and consumption while improving resilience. This work will also be extended to other distribution grids, especially larger ones, in order to extrapolate the results to all distribution grids. In this case, the issue of simulation time arises. The chosen method should be adapted to a statistical approach so as to avoid an exhaustive simulation of the database. Moreover, the influence of agent placement has not been studied; note that a

study of optimal agent placement to maximize the value of ENE will offer an interesting future step.

**Author Contributions:** Methodology, H.S., G.J., R.L.G.L. and H.B.A.; Software, H.S.; Validation, H.S., G.J., R.L.G.L. and H.B.A.; Writing—original draft, H.S.; Writing—review & editing, H.S., G.J., R.L.G.L. and H.B.A.; Supervision, G.J., R.L.G.L. and H.B.A. All authors have read and agreed to the published version of the manuscript.

**Funding:** This research received no external funding.

**Conflicts of Interest:** The authors declare no conflict of interest.

## Abbreviations

The following abbreviations are used in this manuscript:

DSO    Distribution system operator
ENE    Energy not exchanged
nENE   Normalized energy not exchanged
ENS    Energy not supplied
nENS   Normalized energy not supplied
HILP   High-impact low-probability
OPF    Optimal power flow
SPI    Self-producing index
PCC    Point of common coupling

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
