# Peer review of "Energy Not Exchanged: A Metric to Quantify Energy Resilience in Smart Grids"

_sustainability, doi:10.3390/su15032596_

Round 1

Reviewer 1 Report

This paper aims to propose a metric for quantifying smart grid resilience. The problem of defining an all-encompassing, reliable indicator for resilience is a contemporary issue that necessitated the present study. However, the following are significant points that must be addressed;

1. The title is too broad, given that the authors examined only one aspect of assessing resilience. The process of quantifying resilience is a multidimensional topic, as the authors pointed to; consequently, the title must be appropriate to the dimension studied in this research. 

2. What is the difference between the proposed metric and a standard measurement of efficiency? The proposed metric is merely a measurement of the difference between the output and inputs that has been normalized by an standardized value.

3. By using the trapezoidal model, it is evident that the proposed metric does not take into account the time dimension of gauging resilience. The current metric fails to account for the importance of the temporal dimension to the measurement of resilience. This must be accommodated for either in the proposed metric or in the result analyses. 

4. The authors must provide the rate of the metric versus the downtime period as a matter of utmost importance. This will allow readers to comprehend the essence of the simulation. Without these outcomes, the stated results will be difficult to accept.

Author Response

Dear reviewer 1,

Please find attached to this message our answer to your review.

Best regards,

The authors

Reviewer 2 Report

The paper proposed an assessment method for quantifying the resiliency of smart distribution grids. The paper is well-written and in good shape. However, the following concerns should be addressed to improve the quality of this paper.

1-      The main contributions of the paper are unclear to the reviewer. Although the authors present a subsection discussing the contributions of the work, it is not clear that what is new in this subsection.

2-      In section 1.5, it is stated that “only one line will be down at any given time”. However, resiliency study is associated with high impact low probability events (e.g., hurricanes, earthquakes, floods, …) which cause multiple failures. Could you give more clarification on this matter?

3-      The authors proposed the Energy Resiliency Index (ERI) for quantifying the resiliency of the grid in section 2.1. They claim that this index respects all the desirable properties in section 1.3. The authors are advised to give more clarification on this claim.

4-      The authors should illustrate the effectiveness of their proposed resiliency index by comparing their results with other proposed indecies such as EENS.

Author Response

Dear reviewer 2,

Please find attached to this message our anwers to your review.

Best regards,

The authors

Round 2

Reviewer 1 Report

Authors have made effort to improve the original manuscript. It can be accepted for publication. 

Reviewer 2 Report

All my concerns were addressed. No more comments.